# Continuum of care for maternal and newborn health services in Nepal: An analysis from demographic and health survey 2022

**Achyut Raj Pandey**[1,2,*], **Bikram Adhikari**[1], **Raj Kumar Sangroula**[3],
**Parash Mani Sapkota**[1], **Shophika Regmi**[1], **Shreeman Sharma** [1],
**Bishnu Dulal**[1], **Bipul Lamichhane**[1], **Saugat Pratap KC**[1], **Pratistha Dhakal**[1],
**Sushil Chandra Baral**[1]

**1** HERD International, Lalitpur, Nepal, **2** Nepal Health Economics Association, Kathmandu, Nepal, **3** New ERA, Kathmandu, Nepal

* achyutrajpandey2014@gmail.com

## Abstract

### Introduction

With high burden of maternal mortality and stagnant neonatal mortality, maternal and newborn health services have remained a priority program for Nepal. This study aims to assess the determinants of four or more antenatal care (≥4 ANC) visits, institutional delivery (ID), postnatal care (PNC) visit for mother and newborn within the first two days of delivery and the continuum of care.

### Methods

We performed weighted analysis of Nepal Demographic and Health Survey (NDHS) 2022 data accounting for complex survey design. The NDHS is a nationally representative cross-sectional survey that employs a two-stage stratified sampling technique to select participants. We analyzed data from 1,891 women who had live births within two years prior to the survey. Distribution of variables are described using frequency, percentage, and 95% confidence intervals (CI). We performed bivariate and multivariable logistic regression and the results are presented in crude odds ratio (COR), adjusted odds ratio (AOR) and 95% CI.

### Results

In the study, 80.62% (95% CI: 77.95, 83.03) of participants had ≥4 ANC visits, 79.37% (95% CI: 76.68, 81.82) had ID, and 62.56% (95% CI: 56.67, 65.36) received PNC for mother and newborn within two days of delivery. Likewise, 67.59% (95% CI: 64.59, 70.45) had both ≥4 ANC visits and ID, while 51.01% (95% CI: 48.08, 53.93) had all three components of the continuum of care: ≥4 ANC visits, ID, and PNC visit within two days of delivery. The richest wealth quintile participants had three folds higher odds (AOR: 2.98,

**Data availability statement:** Dataset used in this article are available from: https://dhspro-gram.com/data/available-datasets.cfm". Access to the datasets and their supporting documents is granted through a simple process. Individuals need to create an account on the specified portal and explain their purpose. We used individual recode dataset for Nepal (NPIR81FL.DTA of DHS-8) available from the website.

**Funding:** The author(s) received no specific funding for this work.

**Competing interests:** The authors have declared that no competing interests exist.

95% CI: 1.83, 4.83) of completing continuum of care, while the odds were two folds (AOR: 2.04, 95% CI: 1.41, 2.94) higher for richer wealth quintile participants. Participants with birth order three or more had lower odds (AOR: 0.50, 95% CI: 0.36, 0.69) of completing all three continuum of care components. Among other variables associated with continuum of care were province, distance to facility and internet use.

## Conclusion

Significant disparities exist in continuum of care or its components based on wealth quintile, province, and place of residence. Tackling economic gaps, provincial disparities, and leveraging technology are crucial for ensuring fair access to essential maternal health services. Nepal's transition to a federal structure with 7 provinces and 753 local governments with decision making authority presents an opportunity to test and scale up innovative strategies for improving continuum of care coverage.

## Introduction

In 2020, approximately 800 women tragically lost their lives daily, with one death occurring every two minutes adding up to a total of 287,000 death globally due to preventable causes associated with pregnancy and childbirth [1]. Similarly, in 2022, approximately 6,500 newborns lost their lives everyday summing up to a total of 2.3 million global deaths [2]. One third of total neonatal deaths occur on the first day of birth and nearly three-quarters within the first week of life [3]. Neonatal mortality constitutes approximately 47% of all deaths among children under five years of age [2].

In Nepal, maternal mortality ratio (MMR) stands at 151 per 100,000 live births, with 66% of these deaths occurring in postpartum period, 33% during pregnancy, and 6% during delivery [4]. Nepal has been committed to reduce MMR to 70 per 100,000 live births as per Sustainable Development Goals (SDGs) which requires accelerated progress towards coverage of quality maternal health services [4]. Likewise, the present neonatal mortality rate (NMR) is 21 per 1,000 live births [5], and attaining the SDG target of 12 per 1,000 live births by 2030 necessitates an annual rate of reduction (ARR) of 4.8%, surpassing the existing ARR of 4.0% observed from 2000 to 2018 [6,7].

The causes of most of these maternal and neonatal deaths are preventable [1]. Among the leading causes of maternal deaths are non-obstetric complications, obstetric hemorrhage, hypertensive disorder, pregnancy related infections, self-harm, pregnancy with abortive outcomes, and complication of anesthesia management [4]. The leading causes of neonatal death are premature birth, birth complications (such as birth asphyxia or trauma), neonatal infections, and congenital anomalies [8]. In order to achieve a significant reduction in MMR and NMR and progress towards the elimination of preventable causes of maternal and newborn deaths, it is essential to not only expand the reach of healthcare services but also enhance the overall quality of care provided across the entire spectrum of services [9,10]. Maternal mortality and newborn deaths largely stem from inadequate quality care during pregnancies, childbirth and immediately after birth, as well as during the early days of life [1,3,11].

Maternal and newborn health (MNH) service utilization has shown gradual increase in Nepal. Institutional delivery (ID) was 19.0 in 2006 [12], which increased to 43.7% in 2011[13], 64.2% in 2016 [14] and has reached 79.4% in 2022 [5]. The proportion of women receiving any antenatal care (ANC) from a skilled provider increased from 45% in 2006 [12] to 61% in 2011[13], 86% in 2016 [14] and 94% in 2022 [5]. The proportion of women receiving four or

more ANC (≥4 ANC) visits was 31% in 2006 [12], 53% in 2011[13], 71% in 2016 [14], and 81% in 2022 [5]. Similarly, 20% of births in 2006 [12], 44% in 2011 [13], 64% in 2016 [14] and 79% in 2022 [5] occurred in health facilities. In addition, 70% of women received postnatal care (PNC) during first two days of delivery in 2022 [5] which is an increase from 57% in 2016 [14].

Evidence suggest that effective linkage among ANC, delivery care and PNC could improve the health of mother and newborn [15]. Despite policy documents highlighting the importance of a continuum of care [16,17], and an apparent upward trend in the utilization of specific services [5,12–14], research indicates that a significant proportion of women do not complete the continuum of care [18,19]. An analysis of the NDHS 2016 data reveals that only around 41% of women successfully completed all routine maternal care visits, and with notable discontinuation of the service utilization particularly around the time of childbirth [19]. The continuum of care has been emphasized in recent decades as a fundamental principle in maternal, newborn, and child health programs in multiple documents at global level [20–22]. It is considered a crucial strategy for reducing the significant burden of maternal, neonatal and child health problems. Pregnant women must receive ANC integrated with safe childbirth services provided by skilled professionals. PNC is essential for both mothers and newborns during the critical first few weeks after birth, ensuring a seamless connection to newborn care. In cases of complications or illness among women and newborns, ensuring continuity of care—from the household to the hospital, with timely referrals and emergency management—is vital. This continuum of care has also been emphasized in national policy documents such as the Safe Motherhood and Newborn Health (SMNH) Road Map 2030 [17], the ANC and PNC Continuum of Care Guideline [23], and the Nepal's Every Newborn Action Plan (NENAP), 2016 [24]. Having evidence on determinants of continuum of care could be useful from policy and programme perspective in designing maternal and newborn health interventions ensuring that every woman and newborn receive maternal and newborn services as needed in Nepal.

In this context, we identified determinants of ≥4 ANC visits, ID, PNC visit within two days of delivery, combined coverage of ≥4 ANC visits and ID and also the combined coverage of ≥4 ANC visits, ID and PNC visit within two days of delivery.

## Methods

This study involves the analysis of data from the nationally representative Nepal Demographic and Health Survey (NDHS) 2022. The sampling frame used for NDHS 2022 is an updated version of the frame from the Nepal Population and Housing Census (NPHC) 2011, with 36,020 sub-wards which was adjusted for updated urban-rural classification. In NPHC 2011, there were 58 urban municipalities which was increased to 293 urban municipalities by the time of survey, with these urban settings housing around 65% of the population. The NDHS 2022 adopted an updated urban-rural classification system for the survey [5].

### Sampling

Two stage stratified sampling technique was used in NDHS 2022. Seven provinces were divided into urban and rural settings leading to a total of 14 sampling stratum. A strategy of implicit stratification with proportionate allocation was adopted at the lowest administrative levels that included shorting the sampling frame inside each stratum prior to sample selection, including administrative units from various levels, and using a probability-proportional-to-size strategy during the first sampling step. A total of 476 primary sampling units (PSUs) were chosen using a probability-proportional-to-size of the PSU of which 248 PSUs were from urban settings,

while 228 were from rural settings. A household listing was carried out in the selected PSUs, yielding a sampling frame. Segmentation was done in sub-wards when the projected number of homes exceeded 300, and only one segment was selected for the survey. The likelihood of selection was related to the size of the segment. A total of 14,243 households were chosen, and among them, 13,833 were confirmed as being inhabited. From these inhabited households, successful interviews were conducted with 13,786, resulting in a response rate exceeding 99%. All females aged 15 to 49 who were either permanent inhabitants of the designated houses or visitors who had spent the night before in those households, were eligible to participate in the interviews. In this study, we have analyzed the data from 1891 women who had live births two years preceding the survey [5] (Fig 1).

## Data collection tool

Data collection for the survey was conducted between January 5 to June 22, 2022. The DHS Program's model questionnaires were modified to address Nepal's unique demographics and health problems in consultation with various stakeholders, including government departments, agencies, non-governmental organizations, and funders. After the English versions of the survey tools were completed, they were translated into Nepali, Maithili, and Bhojpuri, the three most common languages spoken in Nepal. The translated versions of the questionnaire (household, woman, and man) were then digitized to facilitate data collection using computer-assisted personal interviews (CAPI). Data for the NDHS 2022 was gathered by 19 teams. Each team comprised of a supervisor, one male interviewer, three female interviewers, and a biomarker specialist [5].

## Dependent variables

The dependent variables for this study are maternal health service utilization variables, which includes (i) ≥4 ANC visits, (ii) ID, and (iii) PNC visit for mother and newborn the first two days of delivery, (iv) combined coverage of ≥4 ANC visits and ID, and (v) Continuum of care coverage: combined coverage of ≥4 ANC visits, ID, and PNC visit for mother and newborn during the first two days of delivery. Additional information on variables can be obtained from the full report [5].

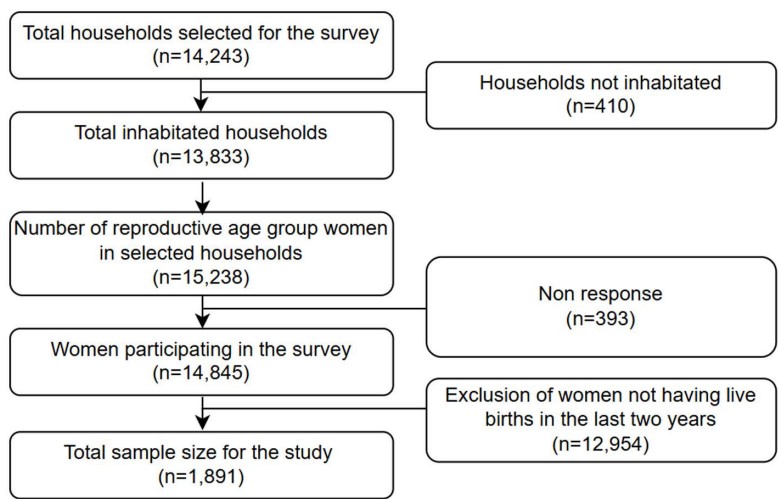

**Fig 1. Sampling process.**

### Independent variables

Independent variables include place of residence (urban, rural), province (Koshi, Madhesh, Bagmati, Gandaki, Lumbini, Karnali, Sudurpashchim), age at delivery (< 20 years, 20-34 years and ≥ 35 years), ethnicity (Brahmin/ Chhetri, Dalit, Janajati, Madhesi, other), wealth quintile (poorest, poorer, middle, richer, richest), education (no education, basic, secondary or higher), distance to facility (< 30 min, 30-59 min, 1-2 hours, and ≥ 2 hours), birth order (one, two, three or more), insurance coverage (no, yes), media exposure-for example newspapers, TV, and radio (no, yes) and internet use (no, yes).

To calculate wealth quintiles, households were assigned scores based on the variety of consumer goods they own, such as televisions, bicycles, or cars, as well as housing characteristics like drinking water sources, toilet facilities, and flooring materials. These scores were derived using principal component analysis. National wealth quintiles were then determined by assigning each household a score, ranking individuals within the household population according to these scores, and dividing the population into five equal groups, with each quintile representing 20% of the population [5].

### Statistical analysis

We conducted data analysis using R version 4.2.0. To accommodate the complex survey design of NDHS 2022, we employed a weighted analysis approach using the "survey" package [25,26]. Multicollinearity was assessed prior to analysis, with a Variance Inflation Factor (VIF) of less than 10 considered an acceptable threshold for multicollinearity. Categorical variables were expressed as frequencies, percentages, and 95% confidence interval (CI). To determine the relationship between independent variables and maternal health service utilization variables (including ≥ 4 ANC visits, ID, and PNC visit, combined coverage of ≥ 4 ANC visits and ID, and combined coverage of ≥ 4 ANC visits, ID and PNC visit within two days of delivery), we performed univariate and multivariable binary logistic regression analyses. The outcomes of the logistic regression analysis were reported as crude odds ratio (COR) and adjusted odds ratio (AOR), each accompanied by their respective 95% CI.

### Ethical consideration

NDHS 2022 obtained ethical approval from the institutional review board of ICF International, United States of America (Reference number: 180657.0.001.NP.DHS.01) and the ethical review board of Nepal Health Research Council (Reference number: 494/2021). Upon approval of our request for access to data, we downloaded NDHS 2022 dataset from https://dhsprogram.com/data/available-datasets.cfm. Written informed consent was obtained in the original survey. In the case of minors, both the assent as well as consent were obtained before enrollment in the survey.

## Results

Out of total participants, 17.37% (95% CI: 15.54, 19.36) of participants at the national level, 15.68% (95% CI: 13.28, 18.42) in urban settings, and 20.59% (95% CI: 18.12, 23.30) in rural settings had age at delivery < 20 years. The highest proportion of participants were Janajati, constituting 30.77% (95% CI: 27.29, 34.48) at the national level, 29.50% (95% CI: 25.10, 34.31) in urban settings, and 33.19% (95% CI: 27.72, 39.15) in rural settings. Regarding education, the proportion of participants having secondary or higher-level education was 47.49% (95% CI: 44.03, 50.97) at the national level, 50.67% (95% CI: 46.02, 55.30) in urban settings, and 41.45% (95% CI: 36.75, 46.31) in rural settings. Approximately 18.49% (95% CI: 15.91, 21.38) of participants at the national level, 17.55% (95% CI: 14.21, 21.47) in urban settings, and

20.28% (95% CI: 16.63, 24.49) in rural settings had no formal education. Additionally, 86.30% (95% CI: 83.55, 88.66) of participants at the national level, 90.67% (95% CI: 87.25, 93.24) in urban settings, and 78.00% (95% CI: 72.85, 82.41) in rural settings reached the nearest health facility within 30 minutes. At the national level, 11.3% (95% CI: 9.23, 13.77) of participants had insurance coverage, 45.76% (95% CI: 42.53, 49.02) had media exposure, and 54.04% (95% CI: 50.72, 57.33) used the internet (Table 1).

Similarly, 79.73% (95% CI: 76.05, 82.98) in urban setting and 82.3% (95% CI: 78.67, 85.43) in rural settings had ≥4 ANC visits, 80.93% (95% CI: 77.32, 84.09) in urban setting and 76.39% (95% CI: 72.34, 80.00) in rural settings had ID, 62.92% (95% CI: 59.08, 66.59) in urban settings and 61.87% (95% CI: 57.60, 65.97) in rural settings had a PNC visit within two days of delivery, 67.98% (95% CI: 64.00, 71.72) in urban settings and 66.83% (95% CI: 62.36, 71.02) in rural settings had both ≥ANC visits and ID, and 51.12% (95% CI: 47.29, 54.93) in urban settings and 50.8% (95% CI: 46.35, 55.25) in rural settings had ≥4 ANC visits, ID, and a PNC visit within two days of delivery (Table 2).

Regarding mothers' age at delivery, 45.88% (95% CI: 39.85, 52.04) of participants with age at delivery <20 years, 52.48% (95% CI: 49.17, 55.77) of participants with age at delivery 20-34 years, and 43.8% (95% CI: 31.1, 57.37) of participants with age at delivery ≥35 years had completed ≥4 ANC visits, had ID, and had PNC visit within two days of delivery. Among ethnic groups, 58.19% (95% CI: 53.26, 62.97) completed ≥4 ANC visits, ID, and PNC visit within two days of delivery among Brahmin/Chhetri, while the proportion was 40.16% (95% CI: 34.07, 46.57) among Madheshi. Among provinces, the percentage of participants completing all three components of the continuum of care was highest in Gandaki at 60.05% (95% CI: 51.93, 67.66) and lowest in Madhesh at 35.95% (95% CI: 30.22, 42.11). While 70.14% (95% CI: 62.97, 76.44) of participants in the richest wealth quintile completed all components of the continuum of care, the proportion was 39.43% (95% CI: 34.85, 44.2) in the poorest wealth quintile. Similarly, there was variation in coverage by educational level, distance to facility, birth order, insurance coverage, media exposure, and internet use, as presented in Table 2.

Compared to participants in Koshi province, participants in Sudurpashchim province had higher odds of completing ≥4 ANC visits (AOR: 2.55, 95% CI: 1.45, 4.49) and had borderline association with ID (AOR: 1.92, 95% CI: 1.03, 3.58). Madhesh province had lower odds of having ID (AOR: 0.43, 95% CI: 0.24, 0.75) and PNC visit (AOR: 0.57, 95% CI: 0.35, 0.93) within two days of delivery. Bagmati province had lower odds (AOR: 0.61, 95% CI: 0.38, 0.98) for completing PNC visit compared to Koshi province. Regarding urban/rural settings, participants in rural setting had marginally higher odds (AOR: 1.50, 95% CI: 1.10, 2.04) of completion of ≥4 ANC visits while variables like ID and PNC visit were not associated (Table 3).

Compared to participants in poorest wealth quintile, participants in richest wealth quintile (AOR: 2.38, 95% CI: 1.09, 5.24) had higher odds of having ≥4 ANC visits. Similarly, compared to poorest wealth quintile, participants in the middle (AOR: 1.78, 95% CI:1.15, 2.75), richer (AOR: 2.22, 95% CI: 1.27, 3.88) and richest (AOR: 9.87, 95% CI: 3.58, 27.20) wealth quintile had higher odds of having ID.

Participants who had secondary or higher education had higher odds of having ID with AOR of 1.90 (95% CI: 1.24, 2.91). Compared to participants who reside at locations with distance to health facility ≥ 2 hours, participants having distance to facility <30 minutes (AOR: 2.17, 95% CI: 1.21, 3.91) and 1-2 hours (AOR: 2.69, 95% CI: 1.34, 5.39) had higher odds of completion of ≥4 ANC visits. Compared to participants with birth order one, participants with birth order three or more had lower odds of completing ≥4 ANC visits (AOR: 0.54, 95% CI: 0.35, 0.83), ID (AOR: 0.27, 95% CI: 0.18, 0.42) and PNC visit (AOR: 0.56, 95% CI: 0.41, 0.76). Among other variables, internet use was associated with completion of ≥4 ANC visits (AOR: 1.61, 95% CI: 1.14, 2.29) and having ID (AOR: 1.46, 95% CI: 1.07, 1.98) (Table 3).

**Table 1. Characteristics of participants.**

| Characteristic | Overall | | Urban | | Rural | |
|---|---|---|---|---|---|---|
| | Weighted n | % (95% CI) | Weighted n | % (95% CI) | Weighted n | % (95% CI) |
| **Age at delivery** | | | | | | |
| <20 years | 328 | 17.37 (15.54, 19.36) | 194 | 15.68 (13.28, 18.42) | 134 | 20.59 (18.12, 23.30) |
| 20–34 years | 1,491 | 78.83 (76.71, 80.81) | 995 | 80.34 (77.40, 82.97) | 495 | 75.97 (73.18, 78.55) |
| ≥35 years | 72 | 3.80 (2.92, 4.93) | 49 | 3.99 (2.86, 5.54) | 22 | 3.44 (2.27, 5.18) |
| **Ethnicity** | | | | | | |
| Brahmin/Chhetri | 484 | 25.62 (22.60, 28.89) | 327 | 26.39 (22.36, 30.87) | 157 | 24.15 (20.20, 28.61) |
| Dalit | 350 | 18.54 (15.82, 21.60) | 221 | 17.87 (14.38, 21.99) | 129 | 19.8 (15.88, 24.41) |
| Janajati | 582 | 30.77 (27.29, 34.48) | 365 | 29.50 (25.10, 34.31) | 216 | 33.19 (27.72, 39.15) |
| Madheshi | 343 | 18.16 (15.17, 21.58) | 239 | 19.29 (15.45, 23.81) | 104 | 16.01 (11.76, 21.44) |
| Others | 131 | 6.92 (4.48, 10.54) | 86 | 6.95 (4.03, 11.73) | 45 | 6.85 (3.30, 13.69) |
| **Province** | | | | | | |
| Koshi | 353 | 18.66 (16.26, 21.32) | 229 | 18.47 (15.51, 21.86) | 124 | 19.02 (15.18, 23.56) |
| Madhesh | 490 | 25.92 (23.13, 28.91) | 361 | 29.15 (25.47, 33.13) | 129 | 19.77 (16.04, 24.12) |
| Bagmati | 288 | 15.24 (12.91, 17.89) | 209 | 16.89 (13.74, 20.59) | 79 | 12.10 (9.38, 15.47) |
| Gandaki | 116 | 6.13 (4.94, 7.58) | 76 | 6.16 (4.57, 8.27) | 39 | 6.05 (4.69, 7.79) |
| Lumbini | 319 | 16.87 (14.99, 18.92) | 179 | 14.48 (12.27, 17.02) | 139 | 21.39 (18.17, 25.01) |
| Karnali | 143 | 7.55 (6.55, 8.69) | 69 | 5.54 (4.56, 6.72) | 74 | 11.37 (9.30, 13.83) |
| Sudurpashchim | 182 | 9.64 (8.399, 11.05) | 115 | 9.30 (7.73, 11.16) | 67 | 10.29 (8.43, 12.49) |
| **Wealth quintile** | | | | | | |
| Poorest | 423 | 22.35 (19.70, 25.25) | 173 | 13.95 (11.25, 17.17) | 250 | 38.33 (32.93, 44.03) |
| Poorer | 429 | 22.67 (19.87, 25.74) | 285 | 22.98 (19.34, 27.08) | 144 | 22.08 (18.11, 26.64) |
| Middle | 370 | 19.56 (17.14, 22.23) | 235 | 18.95 (15.84, 22.50) | 135 | 20.71 (17.15, 24.80) |
| Richer | 375 | 19.83 (17.41, 22.49) | 274 | 22.14 (19.05, 25.58) | 101 | 15.43 (11.86, 19.84) |
| Richest | 295 | 15.59 (13.01, 18.58) | 272 | 21.98 (18.10, 26.42) | 22 | 3.44 (2.27, 5.20) |
| **Education level** | | | | | | |
| No education | 350 | 18.49 (15.91, 21.38) | 217 | 17.55 (14.21, 21.47) | 132 | 20.28 (16.63, 24.49) |
| Basic education | 643 | 34.02 (31.49, 36.65) | 394 | 31.79 (28.47, 35.30) | 249 | 38.27 (34.65, 42.03) |
| Secondary or higher | 898 | 47.49 (44.03, 50.97) | 628 | 50.67 (46.02, 55.30) | 270 | 41.45 (36.75, 46.31) |
| **Distance to health facility** | | | | | | |
| < 30 minutes | 1,632 | 86.30 (83.55, 88.66) | 1,123 | 90.67 (87.25, 93.24) | 508 | 78.00 (72.85, 82.41) |
| 30–59 minutes | 41 | 2.18 (1.61, 2.95) | 17 | 1.40 (0.825, 2.381) | 24 | 3.67 (2.59, 5.17) |
| 1–2 hours | 150 | 7.93 (6.30, 9.93) | 65 | 5.22 (3.476, 7.761) | 85 | 13.07 (10.01, 16.90) |
| ≥2 hours | 68 | 3.59 (2.50, 5.13) | 34 | 2.71 (1.52, 4.79) | 34 | 5.26 (3.40, 8.06) |
| **Birth Order** | | | | | | |
| One | 762 | 40.31 (37.72, 42.95) | 508 | 40.99 (37.56, 44.51) | 254 | 39.02 (35.37, 42.79) |
| Two | 672 | 35.52 (33.05, 38.08) | 453 | 36.6 (33.18, 40.16) | 218 | 33.48 (30.49, 36.61) |
| Three or more | 457 | 24.17 (21.90, 26.59) | 278 | 22.41 (19.65, 25.44) | 179 | 27.51 (23.71, 31.66) |
| **Insurance coverage** | | | | | | |
| No | 1,677 | 88.7 (86.23, 90.77) | 1,079 | 87.09 (83.49, 89.99) | 598 | 91.76 (89.31, 93.69) |
| Yes | 214 | 11.3 (9.23, 13.77) | 160 | 12.91 (10.01, 16.51) | 54 | 8.24 (6.31, 10.69) |
| **Media exposure** | | | | | | |
| No | 1,026 | 54.24 (50.98, 57.47) | 625 | 50.48 (46.05, 54.90) | 400 | 61.4 (57.17, 65.46) |
| Yes | 865 | 45.76 (42.53, 49.02) | 614 | 49.52 (45.10, 53.95) | 252 | 38.60 (34.54, 42.83) |
| **Internet use** | | | | | | |
| No | 869 | 45.96 (42.67, 49.28) | 485 | 39.13 (34.81, 43.62) | 384 | 58.95 (54.60, 63.17) |
| Yes | 1,022 | 54.04 (50.72, 57.33) | 754 | 60.87 (56.38, 65.19) | 268 | 41.05 (36.83, 45.40) |

Note: The number might vary marginally from the total *n* as we have used weighted *n* with rounding off of decimal places.

At national level, 80.62% (95% CI: 77.95, 83.03) participants had at ≥4 ANC visits, 79.37% (95% CI: 76.68, 81.82) had ID, 67.59% (95% CI: 64.59, 70.45) had both ≥4 ANC visits and ID, 62.56% (95% CI: 59.67, 65.36) had PNC visit and 51.01% (95% CI: 48.08, 53.93) had ≥4 ANC visits, ID and PNC visit within two days of delivery (Fig 2).

**Table 2. Coverage level of maternal health services.**

| Characteristics | ≥4 ANC | ID | PNC | ≥4 ANC + ID | ≥4 ANC + ID+PNC |
|---|---|---|---|---|---|
| | % (95%CI) | % (95%CI) | % (95%CI) | % (95%CI) | % (95%CI) |
| **Age at delivery** | | | | | |
| <20 years | 75.12 (69.69, 79.86) | 80.68 (75.22, 85.18) | 61.06 (54.69, 67.07) | 62.32 (56.34, 67.95) | 45.88 (39.85, 52.04) |
| 20–34 years | 82.28 (79.4, 84.83) | 79.12 (76.11, 81.84) | 63.22 (60.08, 66.26) | 68.9 (65.59, 72.03) | 52.48 (49.17, 55.77) |
| ≥35 years | 71.35 (58.74, 81.33) | 78.38 (65.82, 87.22) | 55.61 (43.4, 67.17) | 64.38 (50.15, 76.46) | 43.8 (31.1, 57.37) |
| **Ethnicity** | | | | | |
| Brahmin/Chhetri | 90.44 (87.3, 92.86) | 86.94 (83.06, 90.04) | 66.72 (61.77, 71.33) | 79.3 (75.06, 82.99) | 58.19 (53.26, 62.97) |
| Dalit | 71.37 (64.57, 77.32) | 68.96 (61.92, 75.21) | 55.42 (48.54, 62.1) | 55.7 (49.1, 62.1) | 42.74 (36.26, 49.47) |
| Janajati | 83.88 (79.98, 87.13) | 83.47 (78.97, 87.16) | 66.8 (61.65, 71.57) | 74.01 (68.92, 78.53) | 57.69 (52.5, 62.72) |
| Madheshi | 72.93 (66.35, 78.63) | 75.64 (69.09, 81.19) | 56.84 (49.99, 63.44) | 57.39 (50.55, 63.95) | 40.16 (34.07, 46.57) |
| Others | 74.72 (61.86, 84.34) | 70.7 (55.82, 82.17) | 62.4 (51.56, 72.13) | 54.25 (39.33, 68.44) | 45.32 (33.36, 57.84) |
| **Province** | | | | | |
| Koshi | 78.8 (73.06, 83.59) | 81.96 (75.65, 86.91) | 67.27 (60.11, 73.71) | 66.67 (59.87, 72.85) | 53.16 (45.82, 60.38) |
| Madhesh | 68.85 (61.47, 75.39) | 66.57 (60.57, 72.08) | 52.18 (45.96, 58.34) | 50.22 (43.84, 56.59) | 35.95 (30.22, 42.11) |
| Bagmati | 88.56 (81.32, 93.23) | 88.18 (81.12, 92.83) | 61.37 (52.35, 69.67) | 80.49 (71.77, 87.01) | 57.32 (48.29, 65.9) |
| Gandaki | 84.44 (77.04, 89.78) | 87.58 (80.05, 92.54) | 71.72 (62.3, 79.55) | 76.13 (68.04, 82.69) | 60.05 (51.93, 67.66) |
| Lumbini | 86.84 (81.62, 90.75) | 84.05 (76.54, 89.49) | 70.14 (63.59, 75.96) | 75.14 (67.87, 81.22) | 59.65 (53.43, 65.58) |
| Karnali | 80.39 (74.7, 85.05) | 72.44 (62.99, 80.23) | 54.96 (46.46, 63.17) | 63.83 (55.08, 71.75) | 46.48 (38.71, 54.43) |
| Sudurpashchim | 90.06 (85.1, 93.5) | 86.82 (81.36, 90.86) | 70.07 (63.48, 75.93) | 79.95 (73.61, 85.07) | 60 (53.08, 66.55) |
| **Place of residence** | | | | | |
| Urban | 79.73 (76.05, 82.98) | 80.93 (77.32, 84.09) | 62.92 (59.08, 66.59) | 67.98 (64.00, 71.72) | 51.12 (47.29, 54.93) |
| Rural | 82.30(78.67, 85.43) | 76.39 (72.34, 80.00) | 61.87 (57.60, 65.97) | 66.83 (62.36, 71.02) | 50.8 (46.35, 55.25) |
| **Wealth quintile** | | | | | |
| Poorest | 74.79 (69.76, 79.24) | 66.05 (61.02, 70.75) | 50.25 (44.93, 55.56) | 53.57 (48.5, 58.56) | 39.43 (34.85, 44.2) |
| Poorer | 76.48 (71.17, 81.07) | 72.99 (67.33, 77.99) | 59.68 (53.6, 65.47) | 60.27 (55.01, 65.29) | 44.97 (39.33, 50.74) |
| Middle | 77.74 (72.63, 82.14) | 79.97 (75.09, 84.1) | 62.99 (57.43, 68.22) | 65.11 (59.32, 70.5) | 48.64 (42.96, 54.36) |
| Richer | 85.04 (79.47, 89.3) | 86.77 (81.91, 90.48) | 68.44 (62.3, 73.99) | 76.17 (70.47, 81.07) | 58.25 (52.37, 63.91) |
| Richest | 92.97 (87.64, 96.1) | 97.54 (94.35, 98.95) | 76.38 (69.42, 82.16) | 90.51 (85.05, 94.12) | 70.14 (62.97, 76.44) |
| **Education level** | | | | | |
| No education | 67.65 (60.53, 74.03) | 60.03 (53.66, 66.08) | 50.12 (43.84, 56.39) | 48.57 (42.1, 55.09) | 38.09 (32.29, 44.24) |
| Basic education | 75.93 (72.12, 79.38) | 73.9 (69.62, 77.77) | 57.69 (53.06, 62.18) | 58.56 (54.33, 62.66) | 42.35 (38.03, 46.8) |
| Secondary or higher | 89.02 (86.34, 91.23) | 90.81 (88.41, 92.75) | 70.89 (67.31, 74.22) | 81.46 (78.17, 84.35) | 62.24 (58.4, 65.93) |
| **Distance to health facility** | | | | | |
| < 30 minutes | 81.07 (78.13, 83.7) | 80.81 (77.89, 83.44) | 63.95 (60.81, 66.98) | 68.7 (65.42, 71.8) | 52.02 (48.8, 55.22) |
| 30–59 minutes | 80.57 (65.25, 90.16) | 85.29 (71.79, 92.96) | 76.12 (61.61, 86.36) | 69.9 (54.25, 81.97) | 63 (47.56, 76.16) |
| 1–2 hours | 83.36 (75.83, 88.88) | 74.55 (67.48, 80.52) | 53.26 (44.61, 61.71) | 64.96 (57.63, 71.65) | 43.13 (35.87, 50.69) |
| ≥2 hours | 63.76 (49.49, 75.96) | 51.57 (38.25, 64.66) | 41.36 (29.38, 54.45) | 45.22 (31.78, 59.39) | 36.84 (25.18, 50.28) |
| **Birth Order** | | | | | |
| One | 85.21 (82.28, 87.72) | 90.35 (87.99, 92.29) | 69.68 (65.97, 73.16) | 77.91 (74.58, 80.92) | 58.86 (54.99, 62.62) |
| Two | 82.64 (78.62, 86.04) | 78.19 (73.95, 81.9) | 63.47 (58.65, 68.04) | 68.1 (63.49, 72.38) | 52.79 (47.85, 57.68) |
| Three or more | 69.99 (64.75, 74.75) | 62.77 (57.18, 68.04) | 49.33 (44.19, 54.49) | 49.62 (43.93, 55.31) | 35.29 (30.78, 40.07) |
| **Insurance coverage** | | | | | |
| No | 79.14 (76.2, 81.8) | 77.58 (74.73, 80.19) | 60.98 (57.94, 63.94) | 65.22 (62.04, 68.26) | 48.84 (45.85, 51.84) |
| Yes | 92.24 (87.46, 95.3) | 93.38 (88.65, 96.22) | 74.95 (66.87, 81.6) | 86.19 (80.26, 90.56) | 68.03 (59.79, 75.29) |
| **Media exposure** | | | | | |
| No | 77.25 (73.38, 80.7) | 74.02 (70.22, 77.49) | 59.06 (54.97, 63.02) | 61.86 (57.79, 65.78) | 47.11 (43.25, 51.01) |
| Yes | 84.61 (81.56, 87.24) | 85.7 (82.62, 88.32) | 66.71 (62.79, 70.4) | 74.37 (70.72, 77.72) | 55.62 (51.72, 59.45) |

*(Continued)*

**Table 2.** (Continued)

| Characteristics | ≥4 ANC | ID | PNC | ≥4 ANC + ID | ≥4 ANC + ID+PNC |
|---|---|---|---|---|---|
| | % (95%CI) | % (95%CI) | % (95%CI) | % (95%CI) | % (95%CI) |
| **Internet use** | | | | | |
| No | 73.61 (69.6, 77.26) | 70.81 (66.98, 74.37) | 57 (53.03, 60.88) | 56.5 (52.54, 60.38) | 41.95 (38.28, 45.71) |
| Yes | 86.58 (83.35, 89.27) | 86.64 (83.59, 89.2) | 67.28 (63.34, 71) | 77.01 (73.25, 80.39) | 58.71 (54.55, 62.74) |

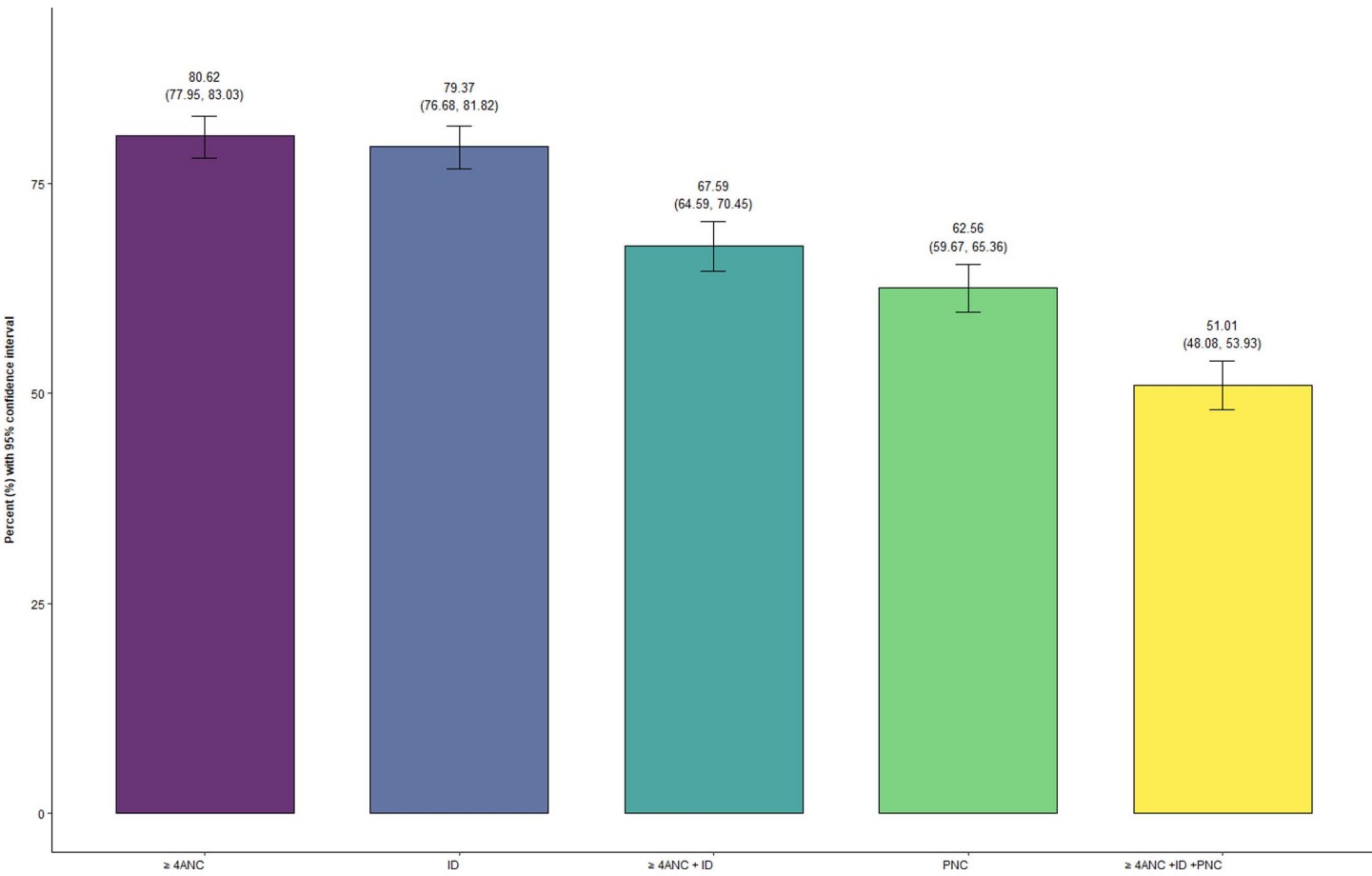

**Fig 2. Maternal and newborn health service utilization and continuum of care.**

Residents of Sudurpashchim province had higher odds of having both ≥4 ANC visits and ID (AOR: 2.69, 95% CI: 1.64, 4.43) and completing all three components of care, i.e., ≥ANC visits + ID + PNC visit (AOR: 1.65, 95% CI: 1.04, 2.61). Similarly, rural residents had marginally higher odds of completing continuum of care compared to urban residents (AOR: 1.33, 95% CI: 1.04, 1.70) (Table 4).

Participants in poorer (AOR: 1.71, 95% CI: 1.23, 2.37), middle (AOR: 1.81, 95% CI: 1.22, 2.69), richer (AOR: 2.65, 95% CI: 1.75, 4.01), and richest (AOR: 5.86, 95% CI: 3.01, 11.40) quintile had higher odds of completing both ≥4 ANC visits and ID. Similarly, participants in poorer (AOR: 1.50, 95% CI: 1.07, 2.12), middle (AOR: 1.54, 95% CI: 1.06, 2.23), richer (AOR: 2.04, 95% CI: 1.41, 2.94), and richest (AOR: 2.98, 95% CI: 1.83, 4.83) quintile had higher odds of completing ≥4 ANC visits, ID and PNC visit within two days of delivery (Table 4).

**Table 3. Factors associated with ≥4 ANC visits, ID and PNC visit within two days.**

| Characteristics | ≥4 ANC | | ID | | PNC | |
|---|---|---|---|---|---|---|
| | COR | AOR | COR | AOR | COR | AOR |
| **Age at delivery** | | | | | | |
| <20 years | Ref | Ref | Ref | Ref | Ref | Ref |
| 20–34 years | 1.54 (1.15, 2.06) | 1.53 (1.02, 2.32) | 0.91 (0.64, 1.28) | 1.16 (0.78, 1.72) | 1.10 (0.83, 1.45) | 1.13 (0.82, 1.55) |
| ≥35 years | 0.82 (0.45, 1.52) | 1.18 (0.57, 2.47) | 0.87 (0.44, 1.72) | 2.08 (0.94, 4.61) | 0.80 (0.46, 1.38) | 1.05 (0.57, 1.94) |
| **Ethnicity** | | | | | | |
| Brahmin/Chhetri | Ref | Ref | Ref | Ref | Ref | Ref |
| Dalit | 0.26 (0.17, 0.41) | 0.54 (0.32, 0.90) | 0.33 (0.22, 0.51) | 0.94 (0.58, 1.54) | 0.62 (0.44, 0.87) | 1.01 (0.69, 1.47) |
| Janajati | 0.55 (0.36, 0.83) | 0.73 (0.46, 1.14) | 0.76 (0.51, 1.13) | 0.91 (0.56, 1.46) | 1.00 (0.74, 1.36) | 1.14 (0.81, 1.60) |
| Madheshi | 0.28 (0.18, 0.45) | 0.59 (0.32, 1.10) | 0.47 (0.30, 0.73) | 1.24 (0.65, 2.36) | 0.66 (0.46, 0.93) | 0.97 (0.59, 1.61) |
| Others | 0.31 (0.16, 0.60) | 0.68 (0.32, 1.46) | 0.36 (0.18, 0.72) | 0.91 (0.45, 1.83) | 0.83 (0.51, 1.33) | 1.29 (0.72, 2.31) |
| **Province** | | | | | | |
| Koshi | Ref | Ref | Ref | Ref | Ref | Ref |
| Madhesh | 0.59 (0.38, 0.93) | 0.81 (0.45, 1.44) | 0.44 (0.28, 0.69) | 0.43 (0.24, 0.75) | 0.53 (0.36, 0.79) | 0.57 (0.35, 0.93) |
| Bagmati | 2.08 (1.09, 3.98) | 1.51 (0.79, 2.88) | 1.64 (0.85, 3.18) | 1.24 (0.65, 2.36) | 0.77 (0.48, 1.24) | 0.61 (0.38, 0.98) |
| Gandaki | 1.46 (0.83, 2.57) | 1.25 (0.68, 2.27) | 1.55 (0.80, 3.03) | 1.22 (0.63, 2.38) | 1.23 (0.73, 2.07) | 1.13 (0.68, 1.86) |
| Lumbini | 1.78 (1.08, 2.93) | 1.70 (1.0, 2.92) | 1.16 (0.64, 2.12) | 1.01 (0.55, 1.85) | 1.14 (0.75, 1.74) | 1.09 (0.71, 1.68) |
| Karnali | 1.10 (0.70, 1.73) | 1.35 (0.79, 2.30) | 0.58 (0.33, 1.02) | 0.87 (0.48, 1.58) | 0.59 (0.38, 0.93) | 0.89 (0.54, 1.45) |
| Sudurpashchim | 2.44 (1.41, 4.22) | 2.55 (1.45, 4.49) | 1.45 (0.84, 2.51) | 1.92 (1.03, 3.58) | 1.14 (0.75, 1.74) | 1.35 (0.83, 2.18) |
| **Place of residence** | | | | | | |
| Urban | Ref | Ref | Ref | Ref | Ref | Ref |
| Rural | 1.18 (0.86, 1.62) | 1.50 (1.10, 2.04) | 0.76 (0.56, 1.03) | 1.08 (0.80, 1.46) | 0.96 (0.75, 1.22) | 1.21 (0.93, 1.57) |
| **Wealth quintile** | | | | | | |
| Poorest | Ref | Ref | Ref | Ref | Ref | Ref |
| Poorer | 1.10 (0.76, 1.58) | 1.28 (0.84, 1.96) | 1.39 (1.00, 1.92) | 1.41 (0.97, 2.04) | 1.47 (1.06, 2.03) | 1.59 (1.10, 2.29) |
| Middle | 1.18 (0.82, 1.69) | 1.14 (0.76, 1.73) | 2.05 (1.46, 2.89) | 1.78 (1.15, 2.75) | 1.69 (1.23, 2.31) | 1.63 (1.11, 2.40) |
| Richer | 1.92 (1.22, 3.00) | 1.48 (0.90, 2.44) | 3.37 (2.19, 5.18) | 2.22 (1.27, 3.88) | 2.15 (1.52, 3.04) | 1.89 (1.25, 2.86) |
| Richest | 4.46 (2.27, 8.74) | 2.38 (1.09, 5.24) | 20.4 (8.51, 49.0) | 9.87 (3.58, 27.20) | 3.20 (2.12, 4.83) | 2.73 (1.59, 4.68) |
| **Education level** | | | | | | |
| No education | Ref | Ref | Ref | Ref | Ref | Ref |
| Basic | 1.51 (1.09, 2.10) | 0.94 (0.65, 1.36) | 1.89 (1.39, 2.55) | 1.05 (0.75, 1.46) | 1.36 (1.01, 1.82) | 0.93 (0.66, 1.31) |
| Secondary or higher | 3.88 (2.65, 5.67) | 1.38 (0.86, 2.22) | 6.58 (4.56, 9.49) | 1.90 (1.24, 2.91) | 2.42 (1.80, 3.26) | 1.23 (0.83, 1.84) |
| **Distance to health facility** | | | | | | |
| ≥2 hours | Ref | Ref | Ref | Ref | Ref | Ref |
| < 30 minutes | 2.43 (1.34, 4.42) | 2.17 (1.21, 3.91) | 3.96 (2.27, 6.90) | 3.15 (1.65, 6.00) | 2.52 (1.48, 4.27) | 1.90 (1.09, 3.29) |
| 30–59 minutes | 2.36 (0.91, 6.11) | 1.97 (0.78, 4.92) | 5.45 (2.08, 14.3) | 5.13 (1.71, 15.4) | 4.52 (1.96, 10.4) | 3.58 (1.61, 7.96) |
| 1–2 hours | 2.85 (1.36, 5.97) | 2.69 (1.34, 5.39) | 2.75 (1.52, 4.98) | 2.95 (1.52, 5.71) | 1.62 (0.93, 2.81) | 1.45 (0.84, 2.52) |
| **Birth Order** | | | | | | |
| One | Ref | Ref | Ref | Ref | Ref | Ref |
| Two | 0.83 (0.61, 1.12) | 0.77 (0.54, 1.08) | 0.38 (0.28, 0.52) | 0.36 (0.25, 0.52) | 0.76 (0.59, 0.97) | 0.78 (0.59, 1.03) |
| Three or more | 0.40 (0.31, 0.53) | 0.54 (0.35, 0.83) | 0.18 (0.13, 0.25) | 0.27 (0.18, 0.42) | 0.42 (0.33, 0.54) | 0.56 (0.41, 0.76) |
| **Insurance coverage** | | | | | | |
| No | Ref | Ref | Ref | Ref | Ref | Ref |
| Yes | 3.13 (1.79, 5.49) | 1.53 (0.85, 2.74) | 4.07 (2.25, 7.39) | 1.48 (0.80, 2.70) | 1.91 (1.27, 2.88) | 1.19 (0.77, 1.84) |
| **Media exposure** | | | | | | |
| No | Ref | Ref | Ref | Ref | Ref | Ref |
| Yes | 1.62 (1.22, 2.14) | 1.05 (0.78, 1.42) | 2.10 (1.59, 2.78) | 1.13 (0.84, 1.52) | 1.39 (1.10, 1.76) | 1.07 (0.83, 1.38) |

*(Continued)*

**Table 3.** (Continued)

| Characteristics | ≥4 ANC | | ID | | PNC | |
|---|---|---|---|---|---|---|
| | COR | AOR | COR | AOR | COR | AOR |
| **Internet use** | | | | | | |
| No | Ref | Ref | Ref | Ref | Ref | Ref |
| Yes | 2.31 (1.70, 3.15) | 1.61 (1.14, 2.29) | 2.67 (2.03, 3.52) | 1.46 (1.07, 1.98) | 1.55 (1.23, 1.95) | 1.05 (0.82, 1.34) |

Odds of completing continuum of care was higher among those who reside 30-59 minutes of travel time to health facility (AOR: 2.24, 95% CI: 1.03, 4.86) compared to those who reside in location with ≥2 hours of travel time. Participants with birth order three or more had lower odds of completing ≥4 ANC visits and ID (AOR: 0.37, 95% CI: 0.25, 0.53) and completing all three continuum of care components (AOR: 0.50, 95% CI: 0.36, 0.69). Participants who used internet had higher odds of completing ≥4 ANC visits and ID (AOR: 1.62, 95% CI: 1.25, 2.11) and complete all three continuum of care components (AOR: 1.38, 95% CI: 1.09, 1.76) (Table 4).

## Discussion

Nepal is committed to achieve SDG target of reducing MMR to 70 per 100,000 live birth and NMR to 12 per 1,000 live births by 2030 [27]. As in some other LMICs, considering the current MMR of 151 per 100,000 live births and NMR of 21 per 1000 live births, achieving SDG target seems challenging without accelerating the service utilization across continuum of care [4,28,29]. Several policy documents in Nepal, such as Nepal Health Sector Strategic Plan 2023-2030 [30], SMNH Roadmap 2030 [17], and ANC and PNC Continuum of Care Guideline [23], and the NENAP 2016 [24], have underscored the importance of ensuring a continuum of care for MNH services for achieving maternal and newborn health related targets. These documents have also outlined corresponding strategies and interventions. Despite these efforts, the stagnant NMR and relatively high MMR indicate the need to further improve service utilization across continuum of care. The aim of this study was to assess the status and explore the factors associated with continuum of care for MNH.

Among the total participants, 80.6% of participants had ≥4 ANC visits which was greater than the coverage level of 53.1% in 2011 [13,27], 70.8% in 2016 [14,31] and 77.9% in 2019 [32]. Similarly, ID rate in our study was 79.4% which was higher than the previous two surveys: 35% in 2011 [13] and 57% in 2016 [14] as per NDHS and 77.5% in 2019 as per Nepal Multiple Indicator Cluster Survey (NMICS) [32]. The coverage of ≥4 ANC visits and ID rates in Nepal are more than some of Southeast Asian countries. For instance, in India, the coverage of ≥4 ANC visits was 57.9% in 2019-21, in Bangladesh, it was 45.8% in 2017-18, and in Pakistan, it stood at 52.8% [31]. Nepal also has higher ID rate than Bangladesh, where the rate was 65% [33], and in Pakistan, where the rate was 66% [34]. However, the ID in this study was lower than that in India where it was 89% [35]. Regarding PNC, 62.56% of the participants had PNC visits for both mother and newborn. In the previous NDHS survey conducted in 2016, 57% of women reported having received a PNC in the first two days of delivery [31]. Similarly, the PNC visit within two days of delivery was 69.4% in the MICS 2019 [32]. The recent DHS survey conducted in Pakistan and Bangladesh have reported lower coverage of PNC compared to Nepal with 60% in Pakistan [34] and 55% in Bangladesh [33]. However, in India, PNC visit in the first two days of delivery was 82% [35] which is higher than this study.

Steady increase in coverage of ≥4 ANC visits, ID and PNC service could be the result of policy commitment Nepal has on MNH services in last few decades. With the overarching goal of enhancing MNH outcomes across the country and to overcome financial barrier in

**Table 4. Factors associated with continuum of care.**

| Characteristics | ≥4 ANC + ID | | ≥4 ANC + ID + PNC | |
|---|---|---|---|---|
| | COR | AOR | COR | AOR |
| **Age at delivery** | | | | |
| <20 years | Ref | Ref | Ref | Ref |
| 20–34 years | 1.34 (1.02, 1.77) | 1.48 (1.03, 2.13) | 1.30 (0.99, 1.71) | 1.29 (0.94, 1.78) |
| ≥35 years | 1.09 (0.58, 2.05) | 1.97 (0.97, 4.02) | 0.92 (0.50, 1.69) | 1.20 (0.63, 2.30) |
| **Ethnicity** | | | | |
| Brahmin/Chhetri | Ref | Ref | Ref | Ref |
| Dalit | 0.33 (0.23, 0.47) | 0.81 (0.53, 1.22) | 0.54 (0.38, 0.75) | 1.03 (0.72, 1.48) |
| Janajati | 0.74 (0.53, 1.03) | 1.00 (0.68, 1.47) | 0.98 (0.74, 1.30) | 1.23 (0.90, 1.69) |
| Madheshi | 0.35 (0.24, 0.51) | 0.79 (0.48, 1.31) | 0.48 (0.35, 0.67) | 0.87 (0.56, 1.35) |
| Others | 0.31 (0.17, 0.58) | 0.66 (0.34, 1.27) | 0.60 (0.35, 1.00) | 1.05 (0.58, 1.90) |
| **Province** | | | | |
| Koshi | Ref | Ref | Ref | Ref |
| Madhesh | 0.50 (0.34, 0.74) | 0.62 (0.38, 1.00) | 0.49 (0.34, 0.73) | 0.58 (0.37, 0.91) |
| Bagmati | 2.06 (1.18, 3.60) | 1.46 (0.86, 2.49) | 1.18 (0.75, 1.88) | 0.88 (0.56, 1.38) |
| Gandaki | 1.59 (0.98, 2.60) | 1.28 (0.75, 2.18) | 1.32 (0.86, 2.05) | 1.14 (0.74, 1.77) |
| Lumbini | 1.51 (0.96, 2.38) | 1.43 (0.90, 2.27) | 1.30 (0.89, 1.91) | 1.27 (0.86, 1.87) |
| Karnali | 0.88 (0.56, 1.40) | 1.47 (0.89, 2.43) | 0.77 (0.50, 1.17) | 1.20 (0.77, 1.89) |
| Sudurpashchim | 1.99 (1.27, 3.14) | 2.69 (1.64, 4.43) | 1.32 (0.88, 1.97) | 1.65 (1.04, 2.61) |
| **Place of residence** | | | | |
| Urban | Ref | Ref | Ref | Ref |
| Rural | 0.95 (0.73, 1.24) | 1.41 (1.08, 1.83) | 0.99 (0.78, 1.25) | 1.33 (1.04, 1.70) |
| **Wealth quintile** | | | | |
| Poorest | Ref | Ref | Ref | Ref |
| Poorer | 1.31 (1.00, 1.73) | 1.71 (1.23, 2.37) | 1.26 (0.93, 1.70) | 1.50 (1.07, 2.12) |
| Middle | 1.62 (1.18, 2.23) | 1.81 (1.22, 2.69) | 1.46 (1.07, 1.98) | 1.54 (1.06, 2.23) |
| Richer | 2.77 (1.96, 3.91) | 2.65 (1.75, 4.01) | 2.14 (1.58, 2.90) | 2.04 (1.41, 2.94) |
| Richest | 8.27 (4.75, 14.4) | 5.86 (3.01, 11.40) | 3.61 (2.48, 5.26) | 2.98 (1.83, 4.83) |
| **Education level** | | | | |
| No education | Ref | Ref | Ref | Ref |
| Basic | 1.50 (1.12, 2.00) | 0.77 (0.55, 1.08) | 1.19 (0.89, 1.61) | 0.69 (0.49, 0.97) |
| Secondary or higher | 4.65 (3.34, 6.48) | 1.27 (0.84, 1.93) | 2.68 (1.99, 3.61) | 1.03 (0.70, 1.53) |
| **Distance to health facility** | | | | |
| ≥ 2 hours | Ref | Ref | Ref | Ref |
| < 30 minutes | 2.66 (1.50, 4.72) | 1.85 (1.00, 3.42) | 1.86 (1.07, 3.22) | 1.28 (0.73, 2.24) |
| 30–59 minutes | 2.81 (1.19, 6.64) | 2.19 (0.92, 5.22) | 2.92 (1.31, 6.52) | 2.24 (1.03, 4.86) |
| 1–2 hours | 2.25 (1.22, 4.14) | 2.09 (1.11, 3.91) | 1.30 (0.72, 2.33) | 1.09 (0.61, 1.94) |
| **Birth Order** | | | | |
| One | Ref | Ref | Ref | Ref |
| Two | 0.61 (0.47, 0.78) | 0.53 (0.38, 0.72) | 0.78 (0.62, 0.99) | 0.77 (0.59, 1.01) |
| Three or more | 0.28 (0.21, 0.37) | 0.37 (0.25, 0.53) | 0.38 (0.30, 0.49) | 0.50 (0.36, 0.69) |
| **Insurance coverage** | | | | |
| No | Ref | Ref | Ref | Ref |
| Yes | 3.33 (2.13, 5.19) | 1.37 (0.85, 2.22) | 2.23 (1.54, 3.24) | 1.25 (0.83, 1.86) |
| **Media exposure** | | | | |
| No | Ref | Ref | Ref | Ref |
| Yes | 1.79 (1.42, 2.26) | 1.06 (0.82, 1.35) | 1.41 (1.14, 1.73) | 1.00 (0.80, 1.25) |

*(Continued)*

**Table 4.** (Continued)

| Characteristics | ≥4 ANC + ID | | ≥4 ANC + ID + PNC | |
|---|---|---|---|---|
| | COR | AOR | COR | AOR |
| **Internet use** | | | | |
| No | Ref | Ref | Ref | Ref |
| Yes | 2.58 (2.02, 3.29) | 1.62 (1.25, 2.11) | 1.97 (1.57, 2.46) | 1.38 (1.09, 1.76) |

service utilization, Nepal initiated maternity incentive scheme in 2005. The scheme initially incentivized transport for facility-based deliveries in 25 districts with low Human Development Index which later evolved into the nationwide Aama Program in 2009, encompassing delivery care. Subsequent policy changes integrated ANC incentives in 2012. The program offers financial incentives to both women and health facilities, with reimbursement rates varying based on the type of service provided and the ecological belt [17,36]. The Maternity incentive scheme serves as a motivating factor for expanding service coverage, benefiting both mothers and health facilities. Furthermore, the free newborn care program was launched which aimed to reduce financial barriers to accessing newborn care services. This incentive-driven approach might have also contributed to overall progress in service utilization.

Although a good proportion of participants completed ≥4 ANC visits, the proportion steadily declined, with only 51% completing all three components of the continuum of care for MNH services. While Nepal government has made significant progress in improving MNH services like ANC visits, ID and PNC checkups, additional efforts are needed to ensure that the chain is maintained across continuum of care. Counselling during 1st ANC visit for the follow up visits, and for ID could be useful strategy which is already reflected in policies of government of Nepal. Ensuring quality of care and gaining trust would require regular supervision and quality assurance activities at health facility level. Establishing inter-institutional linkages between federal, provincial and local government health facilities has the potential to strengthen referral system, enhance the management of complex deliveries and address specific MNH conditions. Such linkages could contribute to saving lives by improving care for both mother and newborn while also fostering greater trust in facilities. Encouraging and supporting Female Community Health Volunteers (FCHVs) in maintaining a list of pregnant women along with their contact information, improving the documentation of personal details in registers, and using mobile phone systems to send appointment reminders to mothers could also improve service utilization across continuum of care.

There are provincial differences in coverage of ≥4 ANC visits, ID, PNC visit, ≥ANC visits + ID and all three components of continuum of care considered in the study. Women residing in Sudurpashchim province had higher odds of completing ≥4 ANC visits, had borderline association with ID and had higher odds completing all three components of care, i.e., ≥ANC visits + ID + PNC visit. Madhesh province had lower odds of having ID and PNC visit within two days of delivery. Variations based on administrative divisions and geographic area were also reported in multiple previous studies in different settings like India [37], Pakistan [38], and Ethiopia [39]. Factors like variation in socio-economic and developmental status, service availability and readiness across facilities in respective region/provinces, quality of the service may be partially responsible for provincial differences in utilization of different components of continuum of care as well as specific service in isolation. Nepal had massive changes in health system with three tiers of governments being functional and local governments taking responsibility of basic health services including MNH services. The transfer of authority to local level governments provides flexibility to tailor interventions or strategies to local needs. This may have some impacts on differential service utilization and coverage status across the

provinces, resulting from innovations tried at local levels. Differences based on provinces and geography could have been observed also because of differences in program implementation strategies, management efficiency and socio-cultural variation among provinces. Additional studies, preferably qualitative, are needed to gain detailed understanding of the reasons causing variation of the service utilization across provinces.

For most services, our study shows disparity in coverage of service based on wealth quintile. Other studies conducted in Ethiopia [40], India [37], and Zambia [41] found that the household wealth quintile was positively associated with ID. Similar to other studies conducted in Nepal [42,43], our study also identified that women of higher wealth quintile were more likely than women with lower wealth quintile to have ID. Similarly, PNC visit within two days of delivery was found lower in poorer wealth quintile in some of the previous studies conducted in Nepal [44] and Ethiopia [45] which align with our study. Having wealth or a higher socio-economic status often means having a higher disposable income, which increases the ability to bear both direct and indirect expenses related to service utilization. Additionally, it provides better access to information and a stronger social network, all of which are positively associated with increased service utilization. Women with high financial standing may have the freedom to decide on the use of household incomes. Persisting disparities in service utilization despite the availability of free maternal health services and incentives raise concern about the effectiveness of the program and point out the need to reevaluate the program. The expansion of community-based services and awareness campaigns, bolstering primary healthcare facilities, and implementation of other measures targeting the economically marginalized are of utmost importance. SMNH roadmap recommends developing a transition plan to revise financial incentives for delivery care and ANC visits with the objective of shifting away from universal incentives to targeting those who currently do not use services, such as the poorest or those in remote areas, while continuing incentives for women in regions with low ANC and ID rates [17]. However, in-depth studies on the potential impact of changes in incentives program is needed before policy or programmatic changes.

Our findings show significant differences in ID based on maternal education level in line with the other studies conducted in Nepal [42,43] and other parts of the world [8,37,40,46]. This could be because education improves women's understanding, facilitates their information access and enables a thorough understanding of advocacy messages communicated through media, the internet, and healthcare providers. Government of Nepal also has the School Health and Nursing Service Program, guided by the National Health Policy, 2019, which has been implemented in 1,011 government schools [36]. Information on adolescent sexual and reproductive health are also covered in the school health program, which could also have served as source of information apart from the content covered in curriculum. Better possession of information about the need of services, where the services are available and that the services are available free of cost with travel incentives may have resulted in better coverage of ID among those who had secondary or higher-level education. However, no association was observed between the completion of ≥4 ANC visits, PNC, and the continuum of care with educational level in adjusted analysis in our study. This could be an area for further research, especially qualitative studies aimed at clearly identifying the factors that limit women's access to these services despite having the necessary information.

Our findings are based on further analysis from nationally representative data collected in globally standardized tool, taking into account the recent federal structure of Nepal. As the survey has used standardized definition for variables, and has used globally accepted survey methodology, findings are comparable with findings from other countries. However, as the survey was undertaken while COVID-19 pandemic was ongoing, there could be some over or underestimation in case of some variables.

## Conclusions

Several factors influenced MNH service utilization and continuum of care, including wealth quintile, education level, place of residence, and internet use. Addressing these disparities is imperative to ensure that all groups of the population can benefit equally from improved MNH services. In addition, establishing inter-linkage between different services and strengthening referral mechanism across different facilities can accelerate progress toward MNH goals, reducing preventable deaths and ensuring the well-being of mothers and newborns. Nepal has recently transitioned to federal structure with 7 provincial and 753 local governments, having decision making authority. This could be an opportunity to test innovative strategies or interventions to increase coverage of continuum of care which can be subsequently scaled up throughout the country.

## Author contributions

**Conceptualization:** Achyut Raj Pandey, Bikram Adhikari.

**Data curation:** Achyut Raj Pandey.

**Formal analysis:** Achyut Raj Pandey, Bikram Adhikari, Parash Mani Sapkota.

**Methodology:** Bikram Adhikari.

**Writing – original draft:** Achyut Raj Pandey, Raj Kumar Sangroula, Shophika Regmi, Shreeman Sharma, Bishnu Dulal, Bipul Lamichhane, Saugat Pratap KC, Pratistha Dhakal, Sushil Chandra Baral.

**Writing – review & editing:** Achyut Raj Pandey, Raj Kumar Sangroula, Parash Mani Sapkota.

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
