## [Decision Letter · Decision Letter 0]

6 Aug 2024

PONE-D-24-08940Determinants of Maternal Health Service Utilization and Continuum of Care in Nepal: An Analysis from Demographic and Health Survey 2022PLOS ONE

Dear Dr. Pandey,

Thank you for submitting your manuscript to PLOS ONE. After careful consideration, we feel that it has merit but does not fully meet PLOS ONE’s publication criteria as it currently stands. Therefore, we invite you to submit a revised version of the manuscript that addresses the points raised during the review process.

We look forward to receiving your revised manuscript.

Kind regards,

Devendra Raj Singh, MSc Public Health, MA

Academic Editor

PLOS ONE

Additional Editor Comments:

Overall, this manuscript presents an important topic and employs appropriate methods for its analysis. However, major revisions are required before being acceptable for publication. In your revision, please consider the suggestions provided by Reviewers for further consideration.

Reviewers' comments:

Reviewer's Responses to Questions

**Comments to the Author**

1. Is the manuscript technically sound, and do the data support the conclusions?

Reviewer #1: Partly

Reviewer #2: Partly

2. Has the statistical analysis been performed appropriately and rigorously? 

Reviewer #1: Yes

Reviewer #2: Yes

3. Have the authors made all data underlying the findings in their manuscript fully available?

Reviewer #1: Yes

Reviewer #2: Yes

4. Is the manuscript presented in an intelligible fashion and written in standard English?

Reviewer #1: No

Reviewer #2: Yes

5. Review Comments to the Author

Reviewer #1: Thank you for providing the opportunity to review the manuscript. The authors have conducted an analysis addressing critical issues, and addressing the gap in the continuum of care is essential for policymakers and program implementers. Once the below comments are addressed, I would recommend this manuscript for publication. However, I suggest reviewing the introduction and discussion sections with experts who have extensive knowledge about the current policies and programs related to maternal and neonatal health.

1. The authors mentioned logistic regression in the abstract, yet Poisson regression is detailed in the methodology section. Additionally, the findings present logistic regression output. Please review the report carefully to ensure coherence.

2. Be consistent in the use of decimal places, either using one or two after the decimal point throughout the document.

3. Regarding the statement, "Madhesh residents had lower odds (AOR: 0.47, 95% CI: 0.23 - 0.99), Sudurpaschim had 27 higher odds (AOR: 2.37, 95% CI: 1.17 - 4.82) for ≥4 ANC visits and health facility delivery than Koshi," I question the significance of the upper limit of 0.99. If rounded to a single decimal place, it may not appear significant. Furthermore, please confirm if the associated p-value is less than 0.05.

It seems like the authors are trying to forcefully emphasize significance.

4. I suggest adding 95% confidence intervals (CI) in the last sentence of the methods section in the abstract. The authors overlooked mentioning CI.

5. In the introduction section of the main body, the authors discuss the SDG target for neonatal mortality. I suggest including a similar discussion for maternal mortality ratio.

6. Consider including information about the policy scenario concerning maternal and neonatal health in Nepal in the introduction section. For example, mention initiatives like the Nepal Safe Motherhood and Newborn Health Road Map 2030 and elaborate on its focus on the continuum of care.

7. It seems unnecessary for authors to use the abbreviation "MCVs" in parentheses (line # 73) since it's not used later in the text.

8. Please ensure the survey year is mentioned in the opening sentence of the methods section.

9. Simplify the discussion of sampling procedures in the methods section. Instead of detailed procedures, provide a brief overview and mention the total number of strata, refer readers to the main report for further details with citations. Additionally, consider including a flowchart illustrating the process leading to the selection of 1933 participants, indicating any missing cases.

10. Clearly define the continuum of care, stating that 'Women who received all three stages of care (Full ANC, ID, and PNC) were considered to have undergone complete Continuum of Care.' Additionally, define Full ANC, ID, and PNC, and clarify how they were calculated, specifying whether women or live births or live births and still births were used as the denominator. Provide rationale for not including stillbirths, if applicable. Alternatively, consider mentioning this information in the sample size flow chart as suggested.

11. Why did not authors include essential variables for the current topic, such as "parity/ or birth order" and "distance to health facility," as covariates.

12. In the independent variables, "age" is mentioned, but in the table, it's labeled as "age at delivery." I suggest maintaining consistency. Additionally, the independent variables do not specify how media exposure, occupation, and health insurance were categorized.

13. Please review and confirm lines 141-143, and cross-check with the results output.

14. Given the smaller sample sizes for rural residence in Table 1, it's important to note that these smaller numbers should be interpreted cautiously to avoid misinterpretation of the results. Consider referencing how the DHS main report addresses and handles smaller sample sizes.

15. Check the spelling of "richer" in Table 2 and ensure that similar minor mistakes are corrected in other tables and categories as well.

16. Consider including "place of residence" in Table 2 and ensure its consistency throughout the table.

17. Consider visualizing the continuum of care for ANC, ANC+ID, and ANC+ID+PNC to enhance understanding.

18. Instead of simply writing "Wealth index combined (Table 2)," suggest using the term "wealth quintile" and providing a brief explanation in the methodology section, referring to the DHS wealth index calculation process.

19. Note that "Internet use" is included in Table 4 but not in Tables 1-3 and is also missing from the list of independent variables.

20. The first paragraph of the discussion should ideally be integrated into the introduction to provide background information for readers.

21. Include a discussion of maternal health inequalities in the introduction and link it to the disparities in continuum of care (CoC) discussed in the findings.

22. Maintain consistency in referring to NMICS or MICS.

23. Improve the discussion by linking evidence of maternal health service inequalities at both national and provincial levels. Consider examining supply-side data to provide further evidence of CoC disparities.

24. Ensure that when short forms like NMICS/MICS are used, they are explained or mentioned in parentheses upon first use to avoid confusion.

25. Overall comment: Check for grammatical errors and consider having the document proofread by native speakers for further refinement.

Reviewer #2: Reviewer Report: Version 1

Date: 4th Aug 2024

Determinants of Maternal Health Service Utilization and Continuum of Care in Nepal:

An Analysis from Demographic and Health Survey 2022

Overall Impression:

Although there are proven evidences of increasing coverage of Antenatal care (ANC), institutional delivery and postnatal care (PNC) could improve the maternal and neonatal health outcomes, little importance has been given to continuum of care (CoC) in Nepal. There are different determinants that drives women for not continuing the services. This manuscript examines the determinants of four or more ANC, institutional delivery, PNC within 2 days of delivery, and also the combined coverage of these variables to assess the determinants of service utilization for CoC in Nepal. Using the recent nationally representative Nepal Demographic and Health Survey 2022, the authors have performed univariate and multivariable regression analyses, the authors have tried to assess the status of continuum of care and explore the factors associated with it for maternal healthcare services. The findings can be important evidence for those who are involved in improving the maternal healthcare service utilization in Nepal. However, the authors lack adequate discussion of policy implication of their findings.

Abstract:

The AOR value for the wealthiest quintile seems unusually high (AOR: 11.96, 95% CI: 14.36 - 32.79). The reported AOR should fall within the range of the confidence interval resulting CI not aligning with AOR. Furthermore, AOR of ~12 indicates that individuals in the wealthiest/richest quintile have almost 12 times higher odds of receiving the combined maternal health services compared to the reference group. Although this is possible, it is unusually high and warrants further scrutiny considering the non-alignment of AOR and CI as mentioned above.

Stating the reference group is missing in some context.

The conclusion is concise and straight forward. It could be further expanded slightly by linking with the policy implications or recommendations based on the findings which are more important for the readers. However, the authors have not discussed the policy implication in their discussion and obviously in the conclusion as well.

Introduction:

The introduction presents the magnitude of the problem of maternal health in global and Nepalese context (para 1). The statistics related to leading cause of maternal mortality and its relation with the quality of healthcare (para 2) and the trend of service utilization (para 3) justifies the motivation of the authors to conduct this study.

Although the authors have mentioned a little about the importance of continuum of care, it would be useful to elaborate. This will allow readers the importance of such intervention that the authors are interested to explore further. Due to the lack of such information, the introduction section has limited theoretical underpinning for conducting this research. Why exploring the determinants of continuum of care is important?

The authors used the term Maternal Care Visits (MCVs) taking the reference #17. Is this term same with ‘Continuum of Care’ that the authors have used throughout this manuscript? What are the similarities and what are the differences of these terms?

Additionally, providing definitions for the variables of interest and explaining how the DHS formulated the survey questions (possibly in a separate table) would enhance the manuscript's clarity. This would ensure that readers understand, for example, that "PNC" refers specifically to "PNC visit for the mother within 2 days of delivery," as used throughout the manuscript.

Method:

The methodology section provides a thorough description of the sampling frame, data collection process, and statistical analysis.

In data collection tools, the authors state that “DHS Program's model questionnaires were modified to address Nepal's unique demographics and health problems in consultation with various stakeholders, including government departments agencies, non-governmental organizations, and funders.” What were the unique demographics and health problems for Nepal identified after such consultation? What were such unique health problems that are relevant for this study? Were the respondents themselves also consulted before modifying the tools? How?

Clarify why only those women who had live births within two years preceding the survey were included in the analysis, and why this specific timeframe was chosen?

This manuscript does not mention how data cleaning was done. Since the authors used the DHS data source, DHS might have explained the data cleaning process elsewhere. A description of how DHS used any methods for cleaning the data would improve the robustness of the methodology.

The independent variables are listed but lack detailed definitions and categories in some places. Providing precise definitions (e.g. basic/secondary/higher education, urban/rural settings) and clarifying the dichotomization of religion variable (into Hindu and others, only in a religiously diverse society which could impact in health service utilization) would enhance further understanding.

Results:

This section requires thorough consistency checks to avoid any similar discrepancies which have occurred in the abstract section earlier.

The overall number does not match with sum of urban and rural. Please check it.

I am not clear why the authors have done subgroup analysis in table 1. The rationale for conducting subgroup analysis might be required.

I would suggest highlighting only key differences or changes the results to avoid repeating the same figures (line 154-157) in

Although the authors have numbered their table, this has not been mentioned in the text. I would suggest for citing table 1, 2, 3 ..... while writing the result section.

Discussion:

The discussion section is well-structured and aligns with the findings. The authors have tried to make a broader interpretation of findings with comparing and contrasting with the previous similar studies or with the previous studies conducted in other countries. The authors have tried to explain why their findings are different if their results conflicts with such compared studies.

The authors argue that federal structure after the political change of 2015 and the transfer of authority to local level governments after the election of 2017 may have led to relatively better planning and effective implementation of interventions which increased the coverage in rural areas at a faster pace than urban areas in 2022 compared to 2016. Again, the authors have discussed the role of FCHVs in encouraging the utilization of MNCH services to increase the utilization in rural areas. Such arguments should be used cautiously. Does this mean, the FCHVs in urban areas (as per the definition of rural areas in DHS survey) have less contribution? To what extent this argument is valid? Is there any possibility that massive increase in number of municipalities/Urban areas after the political changes of 2017 could have resulted this variation? Is the definition of urban areas in 2016 comparable with the definition used in 2022?

The authors stated that they found provincial variations in continuum of care for maternal health services utilization. The authors argue that management approaches and strategies adopted in expanding coverage within the provinces could be responsible for differences. What is the basis under this assumption? What does these management approaches mean here? How these approaches are different by provinces? Can this be clearly stated through the critical review of provincial health plans and policies (if exists)? Is the health plans and policies of Sudhurpaschim province has something different than other provinces which has resulted these findings? Are there specific policies or initiatives in Sudurpaschim that could serve as models for other provinces?

Discussing about the results in discussion section, the manuscript lacks discussion on implications of wide confidence intervals in some cases. Wide CI in some instances suggest there might be significant variability within those subgroups analysis. The discussion of reasons behind such variability example [11.96 (4.36, 32.79) or even 21.15 (8.80, 50.86) in case of richest quintile] and their implication can enhance the manuscript quality.

One of the major drawbacks of this manuscript is it lacks the discussion on policy implication of their findings. For instance,

Age at delivery: I was interested to see the age at delivery for those who were below 20 years. Since the data was collected in 2022 and the survey asked for those who had live births within two years preceding the survey, what does it indicated with the existing laws of National Criminal Codes 2017 that criminalizes the acts of early marriage? How the health programs should be tailored to address this issue among the young population to prevent early marriage?

The use of internet services and health service utilization is interesting. What can be its policy implication? How can the IEC/BCC interventions be revamped considering the increased access to internet services over the period of time?

Government of Nepal has endorsed ANC to PNC Continnum of care Guideline https://fwd.gov.np/wp-content/uploads/2023/06/ANC-to-PNC-continuum-of-care-guideline-4.pdf. How can this findings be linked with the existing guidelines? What need to be added? These kinds of discussion might be required.

These are only some examples. The discussion on separate subheading for policy implication is essential to make the manuscript better.

Conclusion:

As discussed in the abstract section, this conclusion is straight forward which needs further expansion by linking with the policy implications or recommendations based on the findings which are more important for the readers.

6. PLOS authors have the option to publish the peer review history of their article (what does this mean? ). If published, this will include your full peer review and any attached files.

**Do you want your identity to be public for this peer review?** For information about this choice, including consent withdrawal, please see our Privacy Policy .

Reviewer #1: **Yes: ** Kiran Acharya

Reviewer #2: No

---

## [Author Response · Author response to Decision Letter 1]

27 Oct 2024

Thank you for feedback. We have addressed all feedback from reviewers.

Reviewer 1

Reviewer #1: Thank you for providing the opportunity to review the manuscript. The authors have conducted an analysis addressing critical issues, and addressing the gap in the continuum of care is essential for policymakers and program implementers. Once the below comments are addressed, I would recommend this manuscript for publication. However, I suggest reviewing the introduction and discussion sections with experts who have extensive knowledge about the current policies and programs related to maternal and neonatal health.

1. The authors mentioned logistic regression in the abstract, yet Poisson regression is detailed in the methodology section. Additionally, the findings present logistic regression output. Please review the report carefully to ensure coherence.

Response: Thank you for the feedback. We have corrected this part throughout the manuscript.

2. Be consistent in the use of decimal places, either using one or two after the decimal point throughout the document.

Response: Thank you. We have corrected data to two decimal place throughout the manuscript

3. Regarding the statement, "Madhesh residents had lower odds (AOR: 0.47, 95% CI: 0.23 - 0.99), Sudurpaschim had 27 higher odds (AOR: 2.37, 95% CI: 1.17 - 4.82) for ≥4 ANC visits and health facility delivery than Koshi," I question the significance of the upper limit of 0.99. If rounded to a single decimal place, it may not appear significant. Furthermore, please confirm if the associated p-value is less than 0.05.

Response: Thank you for the feedback. We have revised the regression model with additional variables in the model as suggested by reviewer and have opted caution while interpretation of findings with borderline significance.

It seems like the authors are trying to forcefully emphasize significance.

Response: Thank you for feedback. We have cautiously interpreted the findings with borderline significance and have also revised the discussion section.

4. I suggest adding 95% confidence intervals (CI) in the last sentence of the methods section in the abstract. The authors overlooked mentioning CI.

Response: We have corrected the write up.

5. In the introduction section of the main body, the authors discuss the SDG target for neonatal mortality. I suggest including a similar discussion for maternal mortality ratio.

Response: Thank you. We have revised the manuscript accordingly.

6. Consider including information about the policy scenario concerning maternal and neonatal health in Nepal in the introduction section. For example, mention initiatives like the Nepal Safe Motherhood and Newborn Health Road Map 2030 and elaborate on its focus on the continuum of care.

Response: Thank you. Revised accordingly.

7. It seems unnecessary for authors to use the abbreviation "MCVs" in parentheses (line # 73) since it's not used later in the text.

Response: Corrected in the text. We removed the abbreviation in parentheses

8. Please ensure the survey year is mentioned in the opening sentence of the methods section.

Response: Thank you. Corrected accordingly.

9. Simplify the discussion of sampling procedures in the methods section. Instead of detailed procedures, provide a brief overview and mention the total number of strata, refer readers to the main report for further details with citations. Additionally, consider including a flowchart illustrating the process leading to the selection of 1933 participants, indicating any missing cases.

10. Clearly define the continuum of care, stating that 'Women who received all three stages of care (Full ANC, ID, and PNC) were considered to have undergone complete Continuum of Care.' Additionally, define Full ANC, ID, and PNC, and clarify how they were calculated, specifying whether women or live births or live births and still births were used as the denominator. Provide rationale for not including stillbirths, if applicable. Alternatively, consider mentioning this information in the sample size flow chart as suggested.

Response: Thank you for feedback. Updated

11. Why did not authors include essential variables for the current topic, such as "parity/ or birth order" and "distance to health facility," as covariates.

Response: Thank you for feedback. We have revised the regression model adding the variables suggested and have revised the results and discussion section accordingly.

12. In the independent variables, "age" is mentioned, but in the table, it's labeled as "age at delivery." I suggest maintaining consistency. Additionally, the independent variables do not specify how media exposure, occupation, and health insurance were categorized.

Response: We have maintained consistency in tables and manuscript as suggested by reviewers. We have also added the operational definition of media exposure and health insurance.

13. Please review and confirm lines 141-143, and cross-check with the results output.

Response: Thank you for feedback. We have thoroughly revised the results section.

14. Given the smaller sample sizes for rural residence in Table 1, it's important to note that these smaller numbers should be interpreted cautiously to avoid misinterpretation of the results. Consider referencing how the DHS main report addresses and handles smaller sample sizes.

Response: Thank you for feedback. We have cautiously interpreted the findings considering smaller sample size in rural residence.

15. Check the spelling of "richer" in Table 2 and ensure that similar minor mistakes are corrected in other tables and categories as well.

Response: Corrected

16. Consider including "place of residence" in Table 2 and ensure its consistency throughout the table.

Response: Thank you for feedback. Corrected

17. Consider visualizing the continuum of care for ANC, ANC+ID, and ANC+ID+PNC to enhance understanding.

Response: Thank you for feedback. We have added Fig 1 that helps in visualizing ANC, ID, ANC +ID, PNC and ANC+ID+PNC.

18. Instead of simply writing "Wealth index combined (Table 2)," suggest using the term "wealth quintile" and providing a brief explanation in the methodology section, referring to the DHS wealth index calculation process.

Response: Thank you for feedback. We have revised the manuscript and have replaced wealth index combined with wealth quintile. Description has been added in methodology section, also referring DHS wealth index calculation.

19. Note that "Internet use" is included in Table 4 but not in Tables 1-3 and is also missing from the list of independent variables.

Response: Thank you for feedback. We have added Internet use in the list of independent variables

20. The first paragraph of the discussion should ideally be integrated into the introduction to provide background information for readers.

Response: We have revised the first paragraph of discussion with more closer linkage with introduction

21. Include a discussion of maternal health inequalities in the introduction and link it to the disparities in continuum of care (CoC) discussed in the findings.

Response: We have revised introduction section accordingly.

22. Maintain consistency in referring to NMICS or MICS.

Response: Thank you. We have replaced MICS with NMICS in all places.

23. Improve the discussion by linking evidence of maternal health service inequalities at both national and provincial levels. Consider examining supply-side data to provide further evidence of CoC disparities.

Response: Thank you. Discussion part has been revised

24. Ensure that when short forms like NMICS/MICS are used, they are explained or mentioned in parentheses upon first use to avoid confusion.

Response: We have expanded NMICS in first place it appears.

25. Overall comment: Check for grammatical errors and consider having the document proofread by native speakers for further refinement.

Thank you for feedback. We have thoroughly checked the manuscript for grammatical error and assume that it now meets the standards of PLOS ONE

Reviewer 2

Feedback: The findings can be important evidence for those who are involved in improving the maternal healthcare service utilization in Nepal. However, the authors lack adequate discussion of policy implication of their findings.

The AOR value for the wealthiest quintile seems unusually high (AOR: 11.96, 95% CI: 14.36 - 32.79). The reported AOR should fall within the range of the confidence interval resulting CI not aligning with AOR. Furthermore, AOR of ~12 indicates that individuals in the wealthiest/richest quintile have almost 12 times higher odds of receiving the combined maternal health services compared to the reference group. Although this is possible, it is unusually high and warrants further scrutiny considering the non-alignment of AOR and CI as mentioned above.

Response: Thank you for feedback. We have corrected findings in abstracts. We have revised the regression model as per feedback from reviewer 1 and findings now look more precise. We have checked consistency of findings reported throughout the manuscript.

Feedback: Stating the reference group is missing in some context.

Response: Thank you. We have revised.

Feedback: The conclusion is concise and straight forward. It could be further expanded slightly by linking with the policy implications or recommendations based on the findings which are more important for the readers. However, the authors have not discussed the policy implication in their discussion and obviously in the conclusion as well.

Response: Thank you for feedback. We have added policy implication in discussion and conclusion.

Introduction:

Feedback: Although the authors have mentioned a little about the importance of continuum of care, it would be useful to elaborate. This will allow readers the importance of such intervention that the authors are interested to explore further. Due to the lack of such information, the introduction section has limited theoretical underpinning for conducting this research. Why exploring the determinants of continuum of care is important?

Response: Thank you for the feedback. We have added a paragraph in introduction section addressing the feedback.

Feedback: The authors used the term Maternal Care Visits (MCVs) taking the reference #17. Is this term same with ‘Continuum of Care’ that the authors have used throughout this manuscript? What are the similarities and what are the differences of these terms?

Response: Thank you for feedback. We have maintained consistency in using continuum of care throughout the manuscript.

Feedback: Additionally, providing definitions for the variables of interest and explaining how the DHS formulated the survey questions (possibly in a separate table) would enhance the manuscript's clarity. This would ensure that readers understand, for example, that "PNC" refers specifically to "PNC visit for the mother within 2 days of delivery," as used throughout the manuscript.

Response: Thank you for feedback.

Method:

Feedback: The methodology section provides a thorough description of the sampling frame, data collection process, and statistical analysis. In data collection tools, the authors state that “DHS Program's model questionnaires were modified to address Nepal's unique demographics and health problems in consultation with various stakeholders, including government departments agencies, non-governmental organizations, and funders.” What were the unique demographics and health problems for Nepal identified after such consultation? What were such unique health problems that are relevant for this study? Were the respondents themselves also consulted before modifying the tools? How?

Response: The tool was revised considering the Nepalese context like ethnicity, classification of educational level, and type of facilities that are highly contextual. Revision was based on expert consultation.

Feedback: Clarify why only those women who had live births within two years preceding the survey were included in the analysis, and why this specific timeframe was chosen?

Response: This was to align with the standard approach in reporting maternal health service utilization so the findings are comparable. NDHS also reports indicators 2 years proceeding the survey.

Feedback: This manuscript does not mention how data cleaning was done. Since the authors used the DHS data source, DHS might have explained the data cleaning process elsewhere. A description of how DHS used any methods for cleaning the data would improve the robustness of the methodology.

Response: We used clean data from available from DHS site. Not to duplicate information, we have cited main report as needed in methodology part.

Feedback: The independent variables are listed but lack detailed definitions and categories in some places. Providing precise definitions (e.g. basic/secondary/higher education, urban/rural settings) and clarifying the dichotomization of religion variable (into Hindu and others, only in a religiously diverse society which could impact in health service utilization) would enhance further understanding.

Response: Thank you for feedback. Having long list of categories in independent variables may make regression model to perform weak, specially when numbers on specific categories are less. Recategorization and dichotomization has been an standard approach in running the regression models as in the previous literatures. Despite our best interest, we cannot have long list of categories for independent variables because of statistical reasons

Results:

Feedback: This section requires thorough consistency checks to avoid any similar discrepancies which have occurred in the abstract section earlier.

Response: Thank you for feedback. We have thoroughly revised the manuscript

Feedback: The overall number does not match with sum of urban and rural. Please check it.

Response: Findings section has been thoroughly updated including tables.

Feedback: I am not clear why the authors have done subgroup analysis in table 1. The rationale for conducting subgroup analysis might be required.

Response: We do not have subgroup analysis as such. However, data are presented in table 1 disaggregated by urban/rural setting. We assume providing detail data as possible could be informative from policy and programme perspective.

Feedback: I would suggest highlighting only key differences or changes the results to avoid repeating the same figures (line 154-157)

Response: Results section has been thoroughly revised

Feedback: Although the authors have numbered their table, this has not been mentioned in the text. I would suggest for citing table 1, 2, 3 ..... while writing the result section.

Response: Thank you. Table numbers are now quoted in text

Discussion:

Feedback: The authors argue that federal structure after the political change of 2015 and the transfer of authority to local level governments after the election of 2017 may have led to relatively better planning and effective implementation of interventions which increased the coverage in rural areas at a faster pace than urban areas in 2022 compared to 2016. Again, the authors have discussed the role of FCHVs in encouraging the utilization of MNCH services to increase the utilization in rural areas. Such arguments should be used cautiously. Does this mean, the FCHVs in urban areas (as per the definition of rural areas in DHS survey) have less contribution? To what extent this argument is valid? Is there any possibility that massive increase in number of municipalities/Urban areas after the political changes of 2017 could have resulted this variation? Is the definition of urban areas in 2016 comparable with the definition used in 2022?

Response: Thank you for feedback. The writeup was not meant to say FCHVs in rural setting work better than urban or have higher role. However, considering the high FCHV to population ratio, it is often not possible in urban setting to have complete engagement with every households with pregnant women. However, we have revised the manuscript to make sure that it does not communicate the wrong message.

We have updated the regression model as per feedback from reviewer 1. Urban/rural setting

---

## [Decision Letter · Decision Letter 1]

25 Nov 2024

PONE-D-24-08940R1Continuum of Care for Maternal and Newborn Health Services in Nepal: An Analysis from Demographic and Health Survey 2022PLOS ONE

Dear Dr. Pandey,

Thank you for submitting your manuscript to PLOS ONE. After careful consideration, we feel that it has merit but does not fully meet PLOS ONE’s publication criteria as it currently stands. Therefore, we invite you to submit a revised version of the manuscript that addresses the points raised during the review process.

**ACADEMIC EDITOR: **

Thank you for addressing the previous comments. However, there are some minor revisions that need to be addressed before the manuscript can be accepted for publication. Please carefully review and incorporate the additional feedback provided by reviewers. 

We look forward to receiving your revised manuscript.

Kind regards,

Devendra Raj Singh, MSc Public Health, MA

Academic Editor

PLOS ONE

Journal Requirements:

Additional Editor Comments:

Thank you for addressing the previous comments. However, there are some minor revisions that need to be addressed before the manuscript can be accepted for publication. Please carefully review and incorporate the additional feedback provided by reviewers.

Reviewers' comments:

Reviewer's Responses to Questions

**Comments to the Author**

1. If the authors have adequately addressed your comments raised in a previous round of review and you feel that this manuscript is now acceptable for publication, you may indicate that here to bypass the “Comments to the Author” section, enter your conflict of interest statement in the “Confidential to Editor” section, and submit your "Accept" recommendation.

Reviewer #1: All comments have been addressed

Reviewer #2: All comments have been addressed

2. Is the manuscript technically sound, and do the data support the conclusions?

Reviewer #1: Yes

Reviewer #2: No

3. Has the statistical analysis been performed appropriately and rigorously? 

Reviewer #1: Yes

Reviewer #2: No

4. Have the authors made all data underlying the findings in their manuscript fully available?

Reviewer #1: Yes

Reviewer #2: Yes

5. Is the manuscript presented in an intelligible fashion and written in standard English?

Reviewer #1: Yes

Reviewer #2: No

6. Review Comments to the Author

Reviewer #1: Dear Editor,

Thank you for sending the review of the revision. The authors have revised the manuscript, and I am satisfied with the changes. However, there are a few minor comments worth mentioning:

1. Rather than only including the analysis description in the methods section of the abstract, I suggest adding a sentence about the study design and sample size before discussing the analysis.

2. In the list of independent variables, I recommend clarifying media exposure by specifying examples like newspapers, TV, and radio. This will enhance readers' understanding.

3. Since the authors have three outcomes, I suggest replacing 'multivariable' with 'multivariate analysis' throughout the manuscript, where applicable.

4. I recommend not mentioning R Studio as an analysis tool, as it is an environment rather than a statistical tool. If you wish to include it, please also specify its version.

5. I hope the authors checked for multicollinearity before conducting the regression analysis. I suggest mentioning this explicitly, as variables such as age at delivery and birth order could be correlated.

Reviewer #2: Reviewer Report: Version 2

Date: 23rd Nov 2024

Continuum of Care for Maternal and Newborn Health Services in Nepal: An Analysis from 2 Demographic and Health Survey 2022

Overall Impression:

The authors have done excellent work and have addressed most of the concerns raised by the reviewers. However, the manuscript requires a thorough review of data consistency and comprehensive editing especially in findings and discussion to ensure it is ready for publication.

Introduction:

The first paragraph presents the magnitude of problem in global context. It would be beneficial if the authors update this with recent data to ensure that that they are discussing the current situation more accurately.

Para 3: In this para, the authors have presented the leading causes of maternal mortality. For more comprehensive view, it would be better if they present the leading causes for neonatal mortality as well.

Results:

This section requires a thorough consistency check, as there are several issues with data consistency throughout the first paragraph and table-1. The figures reported in Table 1, (the point estimates and their respective confidence intervals), do not align in many cases for both urban and rural groups. For example, the point estimate of 15.68 (95%CI: 18.12 to 23.30) for the urban group and 20.59 (95%CI: 13.28 to 18.42) for the rural group suggest that the reported point estimate does not fall within the given confidence interval. This inconsistency should be addressed. I recommend that Table 1 undergo a thorough review and verification to make sure that all reported values are consistent and correct. Although the authors have stated that they checked the consistency of the findings, my random check shows further discrepancies now that needs to be corrected.

Discussion:

There appears to be some redundancy in the background section. Please review and address this. Additionally, lines 325–328 could be consolidated to avoid repetition.

Both the results and the discussion section need to be revised/edited to improve clarity. This will help the readers understand easily what the authors mean to say.

Line 282-289: I recommend comparing the coverage over time using similar data sources, if available. Moreover, it would be beneficial to compare the findings with more recent data if available. For example, instead of comparing with the 2017-18 DHS data, the authors could consider referencing the Bangladesh Survey 2022 for a more recent statistic.

7. PLOS authors have the option to publish the peer review history of their article (what does this mean? ). If published, this will include your full peer review and any attached files.

**Do you want your identity to be public for this peer review?** For information about this choice, including consent withdrawal, please see our Privacy Policy .

Reviewer #1: No

Reviewer #2: **Yes: ** Geha Nath Khanal

---

## [Author Response · Author response to Decision Letter 2]

29 Dec 2024

Response to reviewers

Feedback: Rather than only including the analysis description in the methods section of the abstract, I suggest adding a sentence about the study design and sample size before discussing the analysis.

Response: Thank you for the feedback. We have revised as suggested

Feedback: In the list of independent variables, I recommend clarifying media exposure by specifying examples like newspapers, TV, and radio. This will enhance readers' understanding.

Response: Thank you for the feedback. We have revised as suggested

Feedback: Since the authors have three outcomes, I suggest replacing 'multivariable' with 'multivariate analysis' throughout the manuscript, where applicable.

Response: Thank you for the feedback. Rather than having all outcome variables in single model, we ran multiple outcome variable with one outcome variable at a time. We prefer to use the term multivariable analysis. This aligns with Essential Medical Statistics by Betty R. Kirkwood, Jonathan A. C. Sterne

Feedback: I recommend not mentioning R Studio as an analysis tool, as it is an environment rather than a statistical tool. If you wish to include it, please also specify its version.

Response: Thank you for the feedback. We have revised as suggested

Feedback: I hope the authors checked for multicollinearity before conducting the regression analysis. I suggest mentioning this explicitly, as variables such as age at delivery and birth order could be correlated.

Response: Thank you for the feedback. We have revised as suggested

Reviewer 2

Feedback: The manuscript requires a thorough review of data consistency and comprehensive editing especially in findings and discussion to ensure it is ready for publication.

Response: Thank you for the feedback. We have thoroughly rechecked the manuscript for consistency of data and revised the manuscript.

Feedback: The first paragraph presents the magnitude of problem in global context. It would be beneficial if the authors update this with recent data to ensure that that they are discussing the current situation more accurately.

Response: Thank you for the feedback. We have revised as suggested

Feedback: Para 3: In this para, the authors have presented the leading causes of maternal mortality. For more comprehensive view, it would be better if they present the leading causes for neonatal mortality as well.

Response: Thank you for the feedback. We have revised as suggested

Feedback: This (result) section requires a thorough consistency check, as there are several issues with data consistency throughout the first paragraph and table-1. The figures reported in Table 1, (the point estimates and their respective confidence intervals), do not align in many cases for both urban and rural groups. For example, the point estimate of 15.68 (95%CI: 18.12 to 23.30) for the urban group and 20.59 (95%CI: 13.28 to 18.42) for the rural group suggest that the reported point estimate does not fall within the given confidence interval. This inconsistency should be addressed. I recommend that Table 1 undergo a thorough review and verification to make sure that all reported values are consistent and correct. Although the authors have stated that they checked the consistency of the findings, my random check shows further discrepancies now that needs to be corrected.

Response: Thank you for the feedback. We apologize for inconsistency of data. We have now revised the table and write up and rechecked other tables too.

Feedback: There appears to be some redundancy in the background section. Please review and address this. Additionally, lines 325–328 could be consolidated to avoid repetition.

Response: Thank you for the feedback. We have revised as suggested

Feedback: Both the results and the discussion section need to be revised/edited to improve clarity. This will help the readers understand easily what the authors mean to say.

Response: Thank you for the feedback. We have revised as suggested

Feedback: Line 282-289: I recommend comparing the coverage over time using similar data sources, if available. Moreover, it would be beneficial to compare the findings with more recent data if available. For example, instead of comparing with the 2017-18 DHS data, the authors could consider referencing the Bangladesh Survey 2022 for a more recent statistic.

Response: Thank you for feedback. We have compared our findings with most recent data from DHS that have comparable methodology across different setting.

---

## [Decision Letter · Decision Letter 2]

27 Jan 2025

Continuum of Care for Maternal and Newborn Health Services in Nepal: An Analysis from Demographic and Health Survey 2022

PONE-D-24-08940R2

Dear Dr. Pandey,

We’re pleased to inform you that your manuscript has been judged scientifically suitable for publication and will be formally accepted for publication once it meets all outstanding technical requirements.

Kind regards,

Devendra Raj Singh, MSc Public Health, MA

Academic Editor

PLOS ONE

Additional Editor Comments (optional):

Reviewers' comments:

Reviewer's Responses to Questions

**Comments to the Author**

1. If the authors have adequately addressed your comments raised in a previous round of review and you feel that this manuscript is now acceptable for publication, you may indicate that here to bypass the “Comments to the Author” section, enter your conflict of interest statement in the “Confidential to Editor” section, and submit your "Accept" recommendation.

Reviewer #1: All comments have been addressed

Reviewer #2: All comments have been addressed

2. Is the manuscript technically sound, and do the data support the conclusions?

Reviewer #1: Yes

Reviewer #2: Yes

3. Has the statistical analysis been performed appropriately and rigorously? 

Reviewer #1: Yes

Reviewer #2: Yes

4. Have the authors made all data underlying the findings in their manuscript fully available?

Reviewer #1: Yes

Reviewer #2: Yes

5. Is the manuscript presented in an intelligible fashion and written in standard English?

Reviewer #1: Yes

Reviewer #2: Yes

6. Review Comments to the Author

Reviewer #1: All comments have been addressed. I would like to congratulate the authors; the paper is now in very good shape.

Reviewer #2: Dear Editor,

There are some very minor comments and some typo that need to be addressed before accepting for publication.

7. PLOS authors have the option to publish the peer review history of their article (what does this mean? ). If published, this will include your full peer review and any attached files.

**Do you want your identity to be public for this peer review?** For information about this choice, including consent withdrawal, please see our Privacy Policy .

Reviewer #1: **Yes: ** Kiran Acharya

Reviewer #2: **Yes: ** Geha Nath Khanal

---

## [Editor Report · Acceptance letter]

PONE-D-24-08940R2

PLOS ONE

Dear Dr. Pandey,

I'm pleased to inform you that your manuscript has been deemed suitable for publication in PLOS ONE. Congratulations! Your manuscript is now being handed over to our production team.

Kind regards,

on behalf of

Mr. Devendra Raj Singh

Academic Editor

PLOS ONE